A new moradisaurine captorhinid reptile (Amniota: Eureptilia) from the upper Permian of India

Reisz Robert R. 1 2
Chatterjee Sankar 3
Modesto Sean P. seanmodesto@yahoo.ca 4
1 International Centre of Future Science, Dinosaur Evolution Research Center, Jilin University , Changchun , China
2 Department of Biology, University of Toronto at Mississauga , Mississauga , Ontario , Canada
3 Department of Geosciences, Texas Tech University , Lubbock , TX , United States of America
4 Department of Biology, Cape Breton University , Sydney , Nova Scotia , Canada
Marsicano Claudia
Electronic publication date: 2024 Nov 14
Publication date: 2024
Volume: 12
Electronic Location ID: e18394
Received 2024 Mar 12; Accepted 2024 Oct 3
Copyright: ©2024 Reisz et al.
Copyright year: 2024
Copyright holder: Reisz et al.
License: This is an open access article distributed under the terms of the Creative Commons Attribution License, which permits unrestricted use, distribution, reproduction and adaptation in any medium and for any purpose provided that it is properly attributed. For attribution, the original author(s), title, publication source (PeerJ) and either DOI or URL of the article must be cited.
License URL: https://creativecommons.org/licenses/by/4.0/

Keywords: Captorhinidae, Moradisaurinae, High-fibre herbivory, Reptilia

Funding: The Natural Sciences and Engineering Research Council (NSERC) of Canada A New Opportunities Fund award from the Canadian Foundation for Innovation The Nova Scotia Research and Innovation Trust Laboratory research was supported by a Discovery Grant from the Natural Sciences and Engineering Research Council (NSERC) of Canada, a New Opportunities Fund award from the Canadian Foundation for Innovation, and a grant from the Nova Scotia Research and Innovation Trust. The funders had no role in study design, data collection and analysis, decision to publish, or preparation of the manuscript.

==============================
Upper Permian rocks of the former supercontinent Gondwana record climax late Paleozoic terrestrial vertebrate faunas that were dominated numerically and ecologically by therapsid synapsids. Older faunal elements of earlier Paleozoic faunas, such as captorhinid reptiles, are rare and scattered components of the first amniote faunas to inhabit high-latitude regions. Here we describe a new genus and species of moradisaurine captorhinid that represents an archaic faunal element of the high-fibre herbivore fauna of the late Permian of what is now peninsular India. The presence of a relatively broad parietal and three rows of conical teeth on the maxilla and the dentary distinguishes Indosauriscus kuttyi gen. et sp. nov. from other moradisaurines. The hypodigm of I. kuttyi comprises skulls that range in length from 39 mm to 54 mm, and high ossification of the braincase elements and well developed skull-roof sutures, indicate that I. kuttyi adults were smaller than those of most moradisaurines. Results of our phylogenetic analyses suggest that moradisaurines, despite appearing first in the paleotropics, dispersed into temperate, high-latitude regions of Pangea early in their evolutionary history. Moradisaurines in dicynodont-dominated faunas, viz. Indosauriscus kuttyi and Gansurhinus naobaogouensis, were the smallest high-fibre herbivores in their respective faunas. This suggests that small body size may have evolved in these moradisaurines as they co-evolved with the more numerous and diverse dicynodont therapsids.

Introduction

Captorhinid reptiles are one of the great success stories of early amniote evolutionary history. These reptiles arose near the end of the Carboniferous Period (Müller & Reisz, 2005; Reisz, Haridy & Müller, 2016), approximately 300 million years ago, and over the next 40+ million years they diversified into a variety of faunivorous, omnivorous, and herbivorous forms (Clark & Carroll, 1973; Dodick & Modesto, 1995; Sumida et al., 2010; Modesto, Scott & Reisz, 2018). The last surviving captorhinids were contemporaneous to, or minor faunal elements of, late Permian terrestrial vertebrate faunas that were dominated by therapsid synapsids (Taquet, 1969; Kutty, 1972; Gaffney & McKenna, 1979; Modesto & Smith, 2001). Over the course of their evolutionary history, captorhinids seemingly dispersed from the crucible of amniote origins in paleotropical Euramerica across the world, with captorhinid remains reported from all continents except Australia and Antarctica (Vjushkov & Chudinov, 1957; Kutty, 1972; Heaton, 1979; Gow, 2000; Reisz, Haridy & Müller, 2016; Cisneros et al., 2020).

One of the hallmarks of evolutionary history of these early reptiles is the acquistion of high-fibre herbivory in the captorhinid clade Moradisaurinae (de Ricqlès & Taquet, 1982; Dodick & Modesto, 1995; Reisz et al., 2011). Phylogenetic studies suggest that moradisaurines arose during the Cisuralian (early Permian) in the paleotropics of what is now North America (Dodick & Modesto, 1995; Modesto et al., 2016), dispersed during by the Guadalupian (middle Permian) throughout what is now Europe and Asia (Vjushkov & Chudinov, 1957; Reisz et al., 2011; Liu, 2023; Matamales-Andreu et al., 2021; Matamales-Andreu et al., 2023), and by the early Lopingian (late Permian) had reached paleo-high-latitude areas in what is now India (Mueller et al., 2007).

The presence of captorhinids in India was reported first by Kutty (1972) in a brief description of a therapsid-dominated fauna from upper Permian rocks of the Pranhita-Godavari Valley. Although this captorhinid material represents the only non-dicynodont member of the Pranhita-Godavari fauna, it generated little interest apart from its biostratigraphic value (Ray, 1999; Ray, 2000; Ray & Bandyopadhyay, 2003), and it was recognized as moradisaurine only relatively recently (Mueller et al., 2007). This material comprises five skulls and associated elements of the anterior part of the postcranial skeleton. In light of recent advancements in our knowledge of the anatomy of moradisaurine captorhinids (Reisz et al., 2011; Modesto, Lamb & Reisz, 2014; Modesto et al., 2019; LeBlanc et al., 2015; Sidor et al., 2022; Liu, 2023; Matamales-Andreu et al., 2023), we describe the Indian moradisaurine in detail here, investigate its phylogenetic relationships, and discuss its implications for moradisaurine evolution.

Materials and Methods

Materials

ISIR 232, ISIR 233, ISIR 234, ISIR 235, and ISIR 236, all reposited in the Geology Museum, Indian Statistical Institute, Kolkata, India, were prepared using standard mechanical methods.

Geological setting

The Gondwana succession in the Pranhita-Godavari Valley is represented by thick sequences of fluvial-lacustrine sediments, ranging in age from Permian to Cretaceous (King, 1881). Ray (1997) studied the depositional systems of the Permo-Triassic rocks and recognized three formations in ascending order: Barakar, Kundaram, and Kamthi. The captorhinid material was recovered from the Kundaram Formation in the northwestern part of the valley (Kutty, 1972) near the village of Golet in Adilabad district, Andhra Pradesh (Fig. 1). The Kundaram Formation is a typical redbed sequence of fluvial deposits consisting of intercalated layers of red mudstone, sandstone, and ferruginous shale. As in all fluvial deposits, the cross-bedded sandstone unit of the Kundaram Formation represents the channel-related facies, whereas the finer siltstone and mudstone units represent the overbank facies accumulated in floodplain environments (Ray, 1997). The ferruginous mudstone units are generally fossiliferous and have yielded exclusively terrestrial amniotes such as dicynodonts. The fossils are mostly found enclosed in hard ironstone nodules scattered on the surface (Kutty, 1972).

Figure 1 Location and geology maps.

(A) Pranhita-Godavari Valley in India. (B) Permian fossil localities of the Kundarum Formation, near the village Golet. (C) Republic of India overview. Redrawn from Ray & Bandyopadhyay (2003).

Nomenclatural act

The electronic version of this article in Portable Document Format (PDF) will represent a published work according to the International Commission on Zoological Nomenclature (ICZN), and hence the new names contained in the electronic version are effectively published under that Code from the electronic edition alone. This published work and the nomenclatural acts it contains have been registered in ZooBank, the online registration system for the ICZN. The ZooBank LSIDs (Life Science Identifiers) can be resolved and the associated information viewed through any standard web browser by appending the LSID to the prefix http://zoobank.org/. Publication LSID is urn:lsid:zoobank.org:pub:47E6FA4C-AF21-40C3-8925-84B3CC374357. The online version of this work is archived and available from the following digital repositories: PeerJ, PubMed Central SCIE and CLOCKSS.

Source of phylogenetic tree

We used the data matrix of Sidor et al. (2022), which is the most recent cladistic evaluation of moradisaurine interrelationships. Mesquite (Maddison & Maddison, 2023) was used to score the Indian moradisaurine for the 76 phylogenetic characters of Sidor et al. (2022). To this we added Albright, Sumida & Jung’s (2021) character 1 (Adult/mature skull length), enumerated here as character 77, and two new characters (78 and 79, see Supplemental File 2). The Mesquite file (available as Supplemental File 1) was then imported into PAUP* (Swofford, 2021) and the matrix was processed by a heuristic search. Multistate taxa were interpreted as uncertainty, starting tree(s) were obtained via stepwise addition, and tree-bisection-reconnection was employed as the branch-swapping algorithm. We ordered seven characters (see Supplemental File 2). Following the suggestion of Rineau et al. (2015) that ordering ‘clinal’ characters increases the resolving power of the analysis. A bootstrap analysis of 1,000 iterations was also conducted, and Bremer decay values were determined by rerunning heuristic searches after adding a single step to the shortest tree, instructing PAUP* to calculate the strict consensus tree for the resulting trees; this procedure was repeated until there was no phylogenetic resolution in the ingroup, and then the number of steps required to collapse each clade was determined by comparing this series of consensus trees with the shortest tree.

Systematic paleontology

EUREPTILIA (Olson, 1947)	
CAPTORHINIDAE (Case, 1911)	
MORADISAURINAE (de Ricqlès & Taquet, 1982)	
Genus Indosauriscus nov.	

urn:lsid:zoobank.org:act:DD1D0B12-FBE7-4A29-9EE7-2C5E772D8C01

Diagnosis. A small species of moradisaurine captorhinid reptile, characterized by the presence of only three rows of conical teeth on the maxilla and the dentary, and a parietal that is relatively broad, with the transverse breadth of the bone measuring three-quarters its anteroposterior length.

Etymology. After the Republic of India; the Ancient Greek σαυ ~ρoς, for ‘lizard’; and the diminutive Latin suffix ‘-iscus’. The gender is masculine.

Indosauriscus kuttyi sp. nov.

urn:lsid:zoobank.org:act:853DC754-A6F3-4AE0-BE95-3C4C40334C36

Holotype. ISIR 233, skull with maxillae and lower jaw attached.

Referred specimens. ISIR 232, partial skull (Kutty, 1972); ISIR 234, skull fragment showing tooth battery; ISIR 235, partial skull and mandible slightly distorted; ISIR 236, skull and mandible with some postcranial elements attached; ISIR 237, snout portion only.

Horizon and locality. Near the village of Golet, Adilabad district, Andhra Pradesh, India (Fig. 1). Kundarum Formation; upper Permian; homotaxial to the Tropidostoma-Gorgonops and/or Cistecephalus assemblage zones of the Karoo Basin, South Africa (Ray, 1999; Smith et al., 2020), which together range in age from 259.3 to 255.2 Ma (Rubidge et al., 2013).

Diagnosis. As for genus, this being the only known species.

Etymology. The specific epithet honours T. S. Kutty, in recognition of his contributions to vertebrate palaeontology.

Description

Indosauriscus kuttyi has the typical captorhinid apomorphies of the procumbent snout with large premaxillary dentition, the transversely expanded temporal region, the large occipital flange of the squamosal, the absence of the tabular bone and the transversely expanded postparietal bone, the greatly reduced supratemporal bone, and the large stapes. The Indian captorhinid shares distinctive ridge-and-pit sculpturing of the skull roof and mandible with well documented taxa such as Captorhinus laticeps, Captorhinus aguti, Labidosaurus hamatus, Labidosaurikos meachami, Moradisaurus grandis, and Tramuntanasaurus tiai (Heaton, 1979; de Ricqlès & Taquet, 1982; Dodick & Modesto, 1995; Modesto et al., 2007; Matamales-Andreu et al., 2023). Assignment to Moradisaurinae is based on the presence of multiple rows of bullet-shaped teeth on the maxilla and the dentary (Reisz et al., 2011). The reduction in palatal dentition, the enlarged contact between the parasphenoid and the stapes, the increased height of the quadratojugal bone, and the contribution of the postparietal bone to the skull roof are all apomorphies shared with Labidosaurikos meachami, Moradisaurus grandis, and/or Rothianiscus multidontus among moradisaurines, and suggest that Indosauriscus is nested more deeply in Moradisaurinae than Captorhinikos valensis and Sumidadectes chozaensis (formerly “Captorhinikos chozaensis”: Jung & Sues, 2024). A reconstruction of the skull of Indosauriscus kuttyi is shown in Fig. 2. It is based mainly on ISIR 233 with supplementary information from ISIR 232.

Figure 2 Reconstruction of the skull of Indosauriscus kuttyi gen. et sp. nov.

(A) Dorsal, (B) ventral , and (C) right lateral views. Abbreviations: ang, angular; bo, basioccipital; d, dentary; f, frontal; j, jugal; l, lacrimal; m, maxilla; n, nasal; p, parietal; pal, palatine; pm, premaxilla; po, postorbital; pp, postparietal; prf, prefrontal; pt, pterygoid; qj, quadratojugal; sm, septomaxilla; sp, splenial; sq, squamosal; st, supratemporal; su, surangular.

Ontogenetic age

The skulls range in length from 39 mm to 54 mm. These skulls are remarkably small for moradisaurines, which regularly exceeded 250 cm in total length, e.g., 265 mm in Rothianiscus multidontus, 410 mm in Moradisaurus grandis, and 280 mm in Labidosaurikos meachami (Olson & Beerbower, 1953; de Ricqlès & Taquet, 1982; Dodick & Modesto, 1995). The absolutely small size of the Indian captorhinids might, therefore, be indicative of a young ontogenetic age. There is, however, a suite of features that support the inference of skeletal maturity. One of these is the level of suture development on the skull roof, particularly the conspicuous interfingering contacts between the frontals and the nasals, between the frontals and the parietals, and between the two frontals in the largest skull, ISIR 233 (skull length of 54 mm), are highly suggestive that this specimen is an adult individual. The second item of evidence for skeletal maturity is the observation of ISIR 233 that the premaxillary process of the nasal extends ventrally along the dorsal process of the premaxilla and forms the anterior margin of the external naris. This distinctive morphology is seen also in large specimens of Captorhinus laticeps (figure 12 in Heaton, 1979) and in Labidosaurus hamatus (figure 3a in Modesto et al., 2007), and we have seen the same in large specimens of Captorhinus sp. from Richards Spur, Oklahoma (R.R.R. and S.P.M., pers. obsv.); we have not found this morphology in smaller, subadult specimens of these species. The presence of this morphology in ISIR 233, the largest specimen in the hypodigm of Indosauriscus kuttyi, is the third feature that indicates that this specimen is skeletally mature. Finally, ISIR 232 preserves a well-ossified braincase with well developed sutures among the opisthotics and the basi- and exoccipitals; the supraoccipital, which is well ossified, is detached from the rest of the braincase, but ISIR 232 has been compressed obliquely, and this element appears that it would fit firmly in its expected position dorsal to the opisthotics and the basi- and exoccipitals. Related to this observation, the recent description of a small skull of M. grandis suggests that the braincase is the last region of the skull to coalesce in that moradisaurine (Sidor et al., 2022). Accordingly, the superb ossification of the braincase of ISIR 233 is another feature that suggests strongly that the largest specimens of Indosauriscus kuttyi are skeletally mature.

Skull and mandible

The premaxillae of ISIR 233 are well preserved and in perfect articulation with neighboring elements (Fig. 3). They show that the premaxilla of Indosauriscus kuttyi is similar to that of Captorhinus aguti and many other captorhinids in exhibiting is anteroventrally-inclined alveolar ridge that houses a conspicuously large tooth in the first tooth position. Three smaller teeth are present posterior to the first tooth. All premaxillary teeth are slightly recurved posteriorly. The dorsal process contacting the nasal bears an acute alary process that is subequal in length to the midline process (Fig. 3D).

Figure 3 Indosauriscus kuttyi. gen. et sp. nov., ISIR 233, holotype.

Photographs and interpretative drawings of partial skull with adducted mandible in (A) dorsal, (B) right lateral, (C) left lateral and (D) anterior views. Abbreviations: ang, angular; d, dentary; f, frontal; j, jugal; l, lacrimal; m, maxilla; n, nasal; p, parietal; pal, palatine; pf, postfrontal; pm, premaxilla; po, postorbital; pp, postparietal; pra, prearticular; prf, prefrontal; ps, parasphenoid; pt, pterygoid; q, quadrate; qj, quadratojugal; sm, septomaxilla; sp, splenial; sq, squamosal; st, supratemporal; su, surangular.

The septomaxilla is an irregularly shaped element that is set into the posterior half of the external naris (Figs. 3B, 3C and 3D). It contacts the lacrimal and the nasal dorsally and the maxilla ventrally, and with the lacrimal and the maxilla forms the septomaxillary foramen (Fig. 3D). The free anterior edge of the septomaxilla is deeply incised, producing a keyhole-shaped profile for the deeper part of the external naris.

The nasal is a slightly curved sheet of bone that forms most of the roof of the snout. In dorsal aspect it is much shorter and wider than the frontal. Anteriorly it forms a blunt snout tip and articulates with the premaxilla with a deeply serrate suture. The anterolateral corner of the bone has a narrow extension that extends directly ventrally anterior to the external naris to the level of the ventral margin of this opening (Figs. 3B, 3C and 3D). Posteriorly and ventrally the nasal contacts the frontal and the lacrimal along broad, interdigitating sutures. The contact with the prefrontal is unusual in that the posterolateral margin of the nasal is deeply incised by an anterior process of the prefrontal.

The lacrimal extends between the external nares and the orbit (Fig. 3). It forms the posterior margin of the former and the antorbital margin of the latter. From its highest point just anterior to the anterolateral corner of the orbit, the lacrimal decreases in height via weakly meandering sutures with the prefrontal and the nasal to the posterolateral corner of the external naris, where it forms a rim for that opening and contacts the septomaxilla slightly deep to the facial surface. The ventral margin of the lacrimal forms a low, sigmoidal contact with the dorsal surface of the maxilla, but upon contacting the jugal the ventral margin of the lacrimal abruptly changes to a straighter oblique contact with the jugal. The lacrimal terminates posteriorly with a short, acute suborbital process. In dorsal aspect the lacrimal has a smooth, sloping visceral surface that extends medially to contact the palatine. This surface is perforated near the rim of the orbit by a pair of lacrimal puncti (Fig. 3A).

The maxilla has a long, low profile in lateral aspect. Anteriorly it extends a narrow, acute process to overlap the alveolar portion of the premaxilla and contributes to the ventral margin of the external naris (Figs. 3B and 3D). Dorsally it features a low, rounded process that forms a sigmoidal contact with the overlying lacrimal. This dorsal process reaches its highest point just anterior to the contact with the jugal, approximately directly dorsal to the sixth or seventh maxillary tooth position. In lateral aspect the maxilla narrows posteriorly to a sharply acute end that terminates at the level of the orbital midpoint (Fig. 3B); in dorsal aspect it is evident that the maxilla actually continues a little farther posteriorly under the palate (Fig. 3A). In lateral aspect the ventral or alveolar margin is a weakly convex edge that extends from the last premaxillary tooth position to the level of the orbital midpoint, and the right maxilla of ISIR 233 appears to have room for twelve tooth positions. A section through the left maxilla of ISIR 233 suggests that there are at least two rows of maxillary teeth (Fig. 3C). ISIR 234 is preserved as a section through the maxillary tooth-plate dentition (Fig. 4A) and confirms that the maxilla of Indosauriscus kuttyi has a multiple rowed region composed of three rows of teeth. As in other moradisaurines, the tooth rows are parallel to the long axis of the tooth plate. The teeth in the multiple rowed (MR) region are subcircular in cross section and conical in profile (e.g., lateral) view. The tooth cross sections in ISIR 234 (Fig. 4A) suggest that there is a gradual increase in the diameter of the MR teeth from anterior to posterior within a tooth row and from lingual-most tooth row to the labial-most tooth row. These are general trends seen also in the MR dentitions of Labidosaurikos meachami and Moradisaurus grandis (Dodick & Modesto, 1995; Modesto et al., 2019).

Figure 4 Indosauriscus kuttyi gen. et sp. nov., ISIR 232 and 234, referred specimens.

(A) Photograph of ISIR 234, maxillary tooth-plate dentition sections in dorsal view; anterior to the top of image. (B) Photograph of lateral aspect of ISIR 232, skull showing left epipterygoid in lateral view and Meckelian foramen of right mandibular ramus. Abbreviations: ep, epipterygoid; mf, Meckelian foramen.

In relative proportions, the prefrontal of Indosauriscus kuttyi is similar to those of Captorhinus aguti, Labidosauriscus richardi, and Captorhinus kierani (e.g., Modesto, 1998; Modesto, Scott & Reisz, 2018; deBraga, Bevitt & Reisz, 2019), but is distinctive in exhibiting a sharp, triangular process anteromedially that incises the the posterolateral corner of the nasal (Fig. 3A). This anteromedial process of the prefrontal resembles a similar extension on the prefrontals of Labidosaurikos meachami and Tramuntanasaurus tiai (Dodick & Modesto, 1995; Matamales-Andreu et al., 2023). The prefrontals of Captorhinus aguti, Captorhinus kierani, Reiszorhinus olsoni, and Labidosauriscus richardi exhibit incipient anteromedial processes (Modesto, 1998; Sumida et al., 2010; Modesto, Scott & Reisz, 2018; deBraga, Bevitt & Reisz, 2019).

The frontals are rectangular sheets that together roof the interorbital portion of the skull table (Fig. 3A). The anterior portion of each frontal, i.e., the part of the frontal that lies medial to the prefrontal, is narrower and slightly longer than the posterior portion, i.e., the part of the frontal that lies directly medial to the postfrontal. With respect to the posterior part, the anterior part of the frontal is relatively longer than that of Captorhinus aguti (Modesto, 1998), but not as long as those in Labidosaurus hamatus and Labidosaurikos meachami (Dodick & Modesto, 1995; Modesto et al., 2007).

The jugal most closely resembles those of Labidosaurus hamatus and Labidosaurikos meachami in exhibiting a relatively deep, wedge-like profile in lateral aspect (Dodick & Modesto, 1995; Modesto et al., 2007) compared to the more slender, slightly crescentic jugals of C. aguti and C. kierani (Modesto, 1998; deBraga, Bevitt & Reisz, 2019). The jugal of Indosauriscus kuttyi is approximately 40% deeper than that of Captorhinus kierani as measured vertically at the posterior end of the jugal-lacrimal suture and compared to the total length of the jugal. The depth of the jugal of of Indosauriscus kuttyi also appears to be increased posteriorly with a ventral extension of the temporal margin, manifesting as a low, convex ridge that extends from a point just anterior to the orbital midpoint posteriorly to a few millimetres of the posterior end of the quadratojugal (Figs. 3B and 3D). This ventral extension of the temporal margin also results in a quadratojugal that is relatively tall anteriorly. In dorsal aspect (Fig. 3A) the jugal can be seen to exhibit a medially-directed alary process, which would have contacted the palatine and the pterygoid. Together with the palatine and the lacrimal, the jugal forms a large, anteroposteriorly elongate suborbital foramen, which is comparable in relative size with the suborbital foramen of Moradisaurus grandis recently described by Sidor et al. (2022).

The quadratojugal is a slightly arched, quadrangular element forming the posterolateral corner of the temporal region (Figs. 3A, 3B and 3D). It is a relatively long, extending slightly farther anteriorly than the squamosal. The quadratojugal is also a relatively tall element for a captorhinid, extending dorsally to a level even with the ventral-most point of the orbit, a distinction that it shares with Labidosaurikos meachami and Rothianiscus multidontus. Part of the height of the quadratojugal can be attributed to the low, rounded ventral extension of the temporal region manifest in both the quadratojugal and the jugal. In ISIR 233 the quadratojugal-jugal suture is longer than the squamosal-jugal suture (Figs. 3B and 3D), but this does not seem the case in ISIR 236, in which the sutures are subequal. The difference in sutural proportions here may represent individual variation or sexual dimorphism; a greater sample size is needed to discern between these two possibilities.

The postorbital forms the posterior-most rim of the orbit and extends occipitally halfway across the temporal region and forms the anterodorsal portion of the temple (Figs. 3A, 3B, and 3D). Medially it contacts the postfrontal anteriorly and the parietal posteriorly via a broadly convex suture. Laterally it overlies the jugal along a shorter convex contact, and posteriorly the postorbital overlaps the anterodorsal corner of the squamosal with a broad tongue of flat bone in the usual captorhinid manner.

In dorsal aspect the postfrontal (Fig. 3) resembles more closely that of Labidosaurikos meachami (Dodick & Modesto, 1995) than other captorhinids, e.g., Captorhinus aguti, Captorhinus kierani, and Labidosaurus hamatus (Modesto, 1998; Modesto et al., 2007; deBraga, Bevitt & Reisz, 2019), in exhibiting a transversely broad posterior process. This transverse expansion of the postfrontal may be related to the relatively greater breadth of the parietal in Indosauriscus kuttyi (see below).

The parietals are large quadrangular bones that form the posterior portion of the skull table (Fig. 3A). Each parietal is relatively wide, with the transverse width ca. 75% the (total) anteroposterior length of the bone. This greatly exceeds the figures for its closest relatives (58%: deBraga, Bevitt & Reisz, 2019), Labidosaurikos meachami (56%: Dodick & Modesto, 1995), and Labidosaurus hamatus (58%: Modesto et al., 2007). The pineal foramen is a subcircular opening that is positioned immediately anterior to the midpoint of the inter-parietal suture. With respect to total parietal length, the anteroposterior diameter of the pineal foramen is the same relative size as that of its close relatives Labidosaurus hamatus and Labidosaurikos meachami (ca. 8% and 9%, respectively: Dodick & Modesto, 1995), compared to the slightly larger openings in Captorhinus magnus and Captorhinus kierani (both ca. 17%: Kissel, Dilkes & Reisz, 2002; deBraga, Bevitt & Reisz, 2019). There are no large pits that exceed in size all other pits in the honeycombing ridge-and-pit ornamentation, as described for the parietals of both Labidosaurus hamatus and Labidosaurikos meachami (Dodick & Modesto, 1995; Modesto et al., 2007).

The supratemporal is a small, asymmetrical splint of bone that spans the boundary between the skull table and the occiput (Fig. 3A). It is angled with respect to the occipital margin such that its long axis is aligned with the pineal foramen. The narrow, occipital part of the supratemporal is nestled between the postparietal medially and the squamosal laterally, whereas the skull-table portion is slightly broader and occupies a small notch in the posterolateral corner of the parietal.

The posterior margin of the skull table is formed by the paired postparietals (Figs. 3A, 3C and 5B). These are most similar to those of Labidosaurikos meachami in having distinct skull-table and occipital regions. As in that moradisaurine, the skull-table region extends the entire transverse breadth of the element and bears ornamentation of ridges and pits that continue posteriorly from the parietal, with which the postparietal shares a broadly meandering suture. At the occipital margin the postparietal drops directly ventrally to form a smoothly surfaced occipital flange. In occipital (posterior) aspect the occipital flange of the postparietal is claw-shaped, deepest medially where it contacted the other postparietal and the supraoccipital and tapering and curving gradually throughout its length to an acute lateral tip that made contact with the squamosal. In its claw-like profile the occipital flange of the postparietal of Indosauriscus kuttyi resembles more the occipital flanges of Captorhinus aguti and Labidosaurus hamatus (Modesto, 1998; Modesto et al., 2007) than the distinctly shelf-like occipital flange of Labidosaurikos meachami (Dodick & Modesto, 1995).

Figure 5 Indosauriscus kuttyi gen. et sp. nov., referred specimen.

Photographs and interpretative drawings of ISIR 232, partial skull with braincase, mandible, and atlas-axis complex in (A) palatal and (B) occipital views; an asterisk (*) indicates the quadrate foramen. Abbreviations: ang, angular; ar, articular; atc, atlantal centrum; ati, atlantal intercentrum; atn, atlantal neural arch; ax, axis vertebra; bo, basioccipital; d, dentary; ex, exoccipital; h, hyoid element; m, maxilla; op, opisthotic; pal, palatine; pop, paroccipital process; pp, postparietal; pra, prearticular; ps, parasphenoid; pt, pterygoid; q, quadrate; qj, quadratojugal; s, stapes; so, supraoccipital; sp, splenial; sq, squamosal; su, surangular; v, vomer; X, vagus foramen.

In dorsal and lateral aspects the squamosal (Figs. 3 and 5B) more closely resembles those of Captorhinus aguti and other non-moradisaurines in featuring a posteroventral-sloping occipital flange; this is in contrast to the condition in Labidosaurikos meachami, in which the occipital flange is orthogonal to the temporal portion of the bone and, as such, is not visible in lateral aspect. The external surface of occipital flange of the squamosal of Indosauriscus kuttyi, like that of the postparietal, is smoothly finished and in strong contrast to the ridge-and-pit ornamentation on the temporal portion of the bone. In lateral aspect the squamosal closely resembles that documented for other small captorhinids (e.g., Modesto, 1998) except that the length of the squamosal-jugal suture is equal to, or slightly shorter than, that between the quadratojugal and the jugal (rather than vice versa), and the observation that the squamosal-quadratojugal suture slopes posteroventrally (rather than parallel to the ventral margin of the posterior cheek).

The palate of ISIR 233 partially preserved and the surface is overprepared, and so the following description of palatal morphology is based on mostly the better preserved ISIR 232 (Fig. 5A). Only the posterior end of the vomer is exposed in both ISIR 233 and ISIR 232 because of occlusion of the mandible. What is visible of this element is not distinguishable from the vomers of other captorhinids such as Captorhinus aguti or Labidosaurus hamatus. In ventral aspect, the palatine resembles most closely that of Labidosaurikos meachami in having a transversely compressed exposure between the pterygoid and the medially expanded dental plate of the maxilla. The ventral surface appears edentulous, but surface detail is not sufficiently well preserved to rule out the possibility that small teeth may have been present posteromedially where there is a slight swelling. The suture between the palatine and the pterygoid is deeply serrate (i.e., in the region where these two elements form a pterygopalatine patch of teeth, as seen in Labidosaurus hamatus and small captorhinids such as Captorhinus aguti). The dorsal surface of the palatine is exposed in ISIR 233 (Fig. 3A), seen through the right orbit, where it is little more that a flat plate of bone that overlies the maxilla and contacts the lacrimal and the jugal laterally. The palatine and the jugal are slightly disarticulated in ISIR 233, but we infer that the two bones formed a relatively large suborbital foramen when reconstructed (see description for the jugal). In general morphology the pterygoid resembles most closely that of Labidosaurikos meachami, except that teeth are present along the medial margin, on the pterygopalatine swelling, and as a narrow cluster along the posteroventral edge of the transverse flange. Most of the teeth are preserved as stumps, but the well-preserved tooth crowns are small, slightly bulbous cones.

The quadrate is present in those skulls that preserve the jaw-joint region but is not well exposed because it is mostly covered by the skull roof, by the adducted mandible, and by braincase elements. The right quadrate of ISIR 232 (Fig. 5) is the best exposed and what can be seen of this bone is not remarkably different from those of other captorhinids.

The left epipterygoid is exposed in lateral aspect in ISIR 232, where is preserved in perfect articulation atop the dorsal edge of the quadrate flange of the left pterygoid (Fig. 4B). It is a triangular structure with a flange-like base and a finger-like dorsal process. Although the anteroventral-most part of the bone is obscured by matrix left to bolster the superficial portions of the left side of the skull, the remainder of the bone resembles more closely the epipterygoid that Heaton (1979, figure 25a) reconstructed for small captorhinids than the relatively robust element preserved in Labidosaurikos meachami (Dodick & Modesto, 1995, figure 10a).

The braincase is best preserved in ISIR 232, in which it is preserved in situ and almost entirely articulated (Fig. 5). The parabasisphenoid closely resembles in morphology and proportions that of Labidosaurikos meachami, particularly in the extensive overlapping contact with the head of the stapes seen in palatal aspect. The parabasisphenoid resembles that of Captorhinus aguti and Captorhinus kierani in exhibiting cristae ventrolaterales that feature sharp, thin ridges that extend from the level of the basipterygoid processes posteriorly to the level where the cristae flatten out and extend laterally to make contact with the stapes (Fig. 5A). Both opisthotics are preserved in ISIR 232. Each consists of a medial, irregularly shaped region from which a relatively short paroccipital process extends laterally. The medial portion of the opisthotic forms with the exoccipital a small vagus foramen (Fig. 5B). The basioccipital is mostly covered by surrounding elements, but what is visible in ventral aspect is not distinguishable from the morphology of this element in other captorhinids. The exoccipitals are preserved in ISIR 232, but each is covered to a different extent by surrounding elements. The left is the better exposed of the two, and shows that the dorsal portion of the exoccipital is crescentic and forms the lateral margin of the foramen magnum. Unlike other captorhinids, however, the dorsal tip of this bone is narrowly acute. The supraoccipital, exposed only in posterior aspect, is typically captorhinid in construction, consisting of a plate-like body from which a median dorsal process and paired dorsolateral processes extend from its dorsal margin. The supraoccipital is deeply incised ventrally where it forms the dorsal margin of the foramen magnum. The stapes is characteristically captorhinid in being composed of a conspicuously large stapedial head, relatively long columella, and short, but distinct, medially curving dorsal process.

In all specimens the upper and lower jaws are tightly occluded to the skull, prohibiting a detailed description of the dorsal part of the mandible (Figs. 3, 4B and 5). What is discernible indicates the general morphology of the mandible of Indosauriscus kuttyi closely resembles that of other small captorhinids, such as Captorhinus aguti and Captorhinus laticeps. The lateral surfaces of the dentary, the surangular, and the angular are ornamented with the same ridge-and-pit sculpturing seen on the skull, and the ventral surfaces of the dentary, the splenial, the angular, the prearticular, and the articular are smooth and devoid of ornamentation. Like the mandibular rami of its close relatives Labidosaurus hamatus and Labidosaurikos meachami, that of Indosauriscus kuttyi is sigmoidal in ventral aspect, and the mandibular ramus is relatively broad transversely (ca. 12% total mandible length, closer to the 14% of Labidosaurus hamatus and Labidosaurikos meachami than more basal captorhinids, e.g., ca. 8% in Captorhinus aguti). A slight lingual swelling of the splenial in both ventral aspect and in the cross section through the left mandibular ramus of ISIR 233 (Fig. 3C) indicates that a lingual shelf (sensu Modesto et al., 2019) is present on the dentary, as in other moradisaurines in which this region of the mandible is preserved. Whereas we have no direct evidence of multiple rows of teeth on the dentary in the available specimens, the presence of a lingual shelf on the dentary is known only in moradisaurines among captorhinids, and it is associated with the presence of a mandibular (dentary) tooth plate. Considering that the dentary tooth plate exhibits one fewer row of teeth than the corresponding upper tooth plate (e.g., Labidosaurikos meachami: Dodick & Modesto, 1995), we estimate that the dentary supports two rows of teeth.

The splenial and the post-dentary bones, apart from the coronoid, which is not exposed in any of the specimens available to us, resemble in general morphology those of other captorhinids. The only remarkable aspect is the slit-like posterior Meckelian foramen (“foramen intermandibularis caudalis” of Heaton, 1979), seen in the medial view of the right mandibular ramus of ISIR 232 (Fig. 5B): the height of the foramen is only about 15% of its length. Otherwise, the posterior Meckelian foramen appears normally developed, being roughly 7.7% the length of the mandible ramus, which in relative terms is slightly shorter than those of Labidosaurikos meachami and Moradisaurus grandis, in which the same figures are 10.4% and 11.6% (Dodick & Modesto, 1995; Modesto et al., 2019).

A single hyoid element, the first ceratobranchial, is preserved in ISIR 232 (Fig. 5A). It is a relatively elongate, slightly curved rod of bone with a spatulate anterior end. The head is slightly smaller and the shaft is slightly longer than the first ceratobranchial of Captorhinus laticeps (figure 12 in Heaton, 1979).

Postcranial skeleton

ISIR 236 is the only specimen with preserved postcrania (Fig. 6). In dorsal aspect there are two large flat bones that may be scapulocoracoids and what appears to be the proximal end of the left humerus, but the margins of the former are broken along the edges of the specimen, and the surface preservation of all these elements is poor (Fig. 6A). In ventral aspect the preservation is much better and the dermal elements of the pectoral girdle are present in perfect articulation, although smaller disarticulated elements obscure details of the margins of these bones (Fig. 6B). The ventral portions of the clavicles are bifurcated as in those of the small captorhinids (Holmes, 1977), and the anterior, larger process of each clavicle approaches the midline along the anterior margin of the interclavicle; overlying material precludes confident determination whether the clavicles actually make contact with each other medially as they do in small captorhinids (Holmes, 1977). The tongue-like posterior process of the main plate of the clavicle is relatively much larger than in Captorhinus aguti (figure 2d in Holmes, 1977). The greater part of the right humerus is preserved in association with the pectoral girdle. It is preserved in anterodorsal view, with the deltopectoral crest oriented posteriorly. The left humerus distal to the origin for the supinator, across to the entepicondylar foramen, is missing. Numerous disarticulated, vertebral elements and ribs are preserved on and around the pectoral bones, and what is visible of these conforms with general captorhinid morphology and is not otherwise remarkable.

Figure 6 Indosauriscus kuttyi gen. et sp. nov., ISIR 236, referred specimen.

Skull and postcrania in (A) dorsal and (B) ventral views.

Phylogenetic analysis

The phylogeny of captorhinid reptiles is complicated by the inbalance between the number of taxa that have been described and the available potential synapomorphies that can be used to determine patterns of relationships. The majority of taxa are based on cranial materials or fragmentary cranial remains, with differences in dental anatomy and size distinguishing the various species of Captorhinus. Captorhinids therefore appear to be notoriously conservative in their anatomy in this regions of the skeleton, with dental anatomy and the evolution of multiple rows of teeth weighing heavily on the overall list of characters. In addition, size appears to be a major factor, with smaller taxa being more slender and gracile, while younger and larger members being stocky and massive postcranially. Overall, there are more than 25 named taxa, and only 79 characters to evaluate their interrelationships. Of these, 16 characters are related to dentition and only 10 are available postcranially. Excluding some of the poorly known taxa, such as Hecatogomphius and Riabininus, both known only from dentigerous bones (Vjushkov & Chudinov, 1957; Ivakhnenko, 1990), does relatively little to help resolve the pattern of relationships. Our phylogenetic analysis of 24 captorhinid taxa with Protorothyris and Paleothyris as outgroups yielded 4 equally parsimonious trees, each with a tree length of 236 steps (CI 0.445; RI 0.689; HI 0.555). The strict consensus of these trees is shown in Fig. 7. Labidosaurus forms a sister-group relationship with a monophyletic Moradisaurinae, which includes Rhodotheratus parvus as its basal-most member. Indosauriscus is nested well within Moradisaurinae, as the sister taxon of the clade that includes Captorhinikos valensis, Gansurhinus naobaogouensis, Labidosaurikos meachami, Moradisaurus grandis, Rothianiscus multidontus, and Sumidadectes chozaensis. A reduced phylogenetic analysis focusing mainly on Moradisaurinae by removing Paleothyris, Protorothyris, Thuringothyris, Euconcordia, Opisthodontosaurus, Reiszorhinus, Rhodotheratus, Romeria, Protocaptorhinus, and Captorhinus laticeps, and using Rhiodenticulatus as the outgroup, yielded a single most parsimonious tree (heuristic search with 1,000 iterations, 137 steps long, CI = 0.613, RI = 0.635, RC = 0.389), which recovered the same moradisaurine clades as in the full analysis and further resolved Gansurhinus naobaogouensis as the sister of the clade of Labidosaurikos meachami, Moradisaurus grandis, Rothianiscus multidontus, and Sumidadectes chozaensis.

Figure 7 Moradisaurine captorhinid interrelationships.

(A) Strict consensus of four trees following phylogenetic analysis of a data matrix of 24 taxa and 79 characters; moradisaurine branches in green; basal captorhinid branches in blue; outgroup branches in black. (B) Single most parsimonious tree following phylogenetic analysis of reduced number of taxa, with Rhiodenticulatus heatoni as the outgroup; basal captorhinid branches in blue; moradisaurine branches in green. Bremer support (Roman font, above branch) and bootstrap support (italicized font, below branch) values are shown for clades. (C) Resolved moradisaurine topology from tree in B illustrating one biogeographic scenario of two independent dispersals into tropical western Euramerica; branches of North American moradisaurines in open green; branches of extra-North American moradisaurines in solid green. Arrow 1: dispersal of ancestor of Labidosaurikos meachami and Rothianiscus multidontus into tropical western Euramerica (i.e., North America [NA]) from extra-NA Pangea. Arrow 2: dispersal of ancestor of Captorhinikos valensis and Sumidadectes chozaensis into NA from extra-NA Pangea. (D) Resolved moradisaurine topology from tree in B illustrating alternate biogeographic scenario; branch colors as in C. Arrow 3: dispersal of ancestor of clade containing Gansurhinus naobaogouensis, L. meachami, R. multidontus, C. valensis and S. chozaensis into NA from extra-NA Pangea. Arrow 4: dispersal of G. naobaogouensis into extra-NA Pangea from NA.

Discussion

Paleogeographic distribution of captorhinids and moradisaurines within a phylogenetic framework

The evolutionary history of captorhinids is characterized by an early diversification in the equatorial region of Pangea during the late Carboniferous and early Permian. This diversification includes the likely acquisition of omnivory and herbivory, with associated modifications largely in their dentition for effective oral processing of food and increase in size. Increase in size occurred predominantly within the moradisaurine sub-clade of captorhinids (although Reiszorhinus and Labidosaurus also reached moderate to large size among non-moradisaurine captorhinids) all within the early Permian equatorial region of Pangea. Overall, the weakness of the phylogeny of Captorhinidae in general and Moradisaurinae in particular makes detailed assessments about patterns of paleogeographic dispersal of the latter group of reptiles somewhat problematic. The well supported sister-group relationship between Moradisaurinae and the North American Labidosaurus hamatus, and the basal position of the North American Rhodotheratus parvus, together support the hypothesis that moradisaurines originated in western Laurasia during the early Permian. According to our tree topology, moradisaurines achieved a wider geographic distribution early in their evolution but these early diverging moradisaurines do not appear until the end of the middle Permian. Thus, there is some basis for an ‘out of equatorial Pangea’ expansion into middle and late Permian temperate regions including Gondwana, Russia, and China. Interestingly, this dispersal occurred not only among moradisaurines, but there were also separate dispersal events associated with more basal members of the clade. Superficially, the multiple out-of-tropics expansions with no apparent later migration to the equatorial belt conforms to a pattern of biotic diversification by taxon pulses (Erwin, 1979; Folinsbee & Brooks, 2007). There is no evidence that captorhinid lineages ever migrated back into the equatorial belt of Pangea because the available temporal information indicates that the western Laurasian Labidosaurikos is stratigraphically one of the oldest moradisaurines.

The tree topology resulting from our reduced phylogenetic analysis does not allow a decisive reconstruction of the pattern of moradisaurine dispersal out of western Laurasia. This is partly because the ancestor of the clade that contains Indosauriscus and Labidosaurikos could have dispersed to eastern Laurasia and Gondwana, but despite the early appearance of the latter it is possible that the ancestor of Labidosaurikos and its sister taxon Rothianiscus do not represent an early diversification of moradisaurines, but rather a secondary dispersal back to western Laurasia. Nevertheless, if the temporal pattern of the fossil record is taken into account, then the ancestors of the two clades, one containing Moradisaurus and Gansurhinus and the other containing Indosauriscus and Tramuntanasaurus, likely represent separate dispersal events out of western Laurasia. Our current knowledge of the fossil record is problematic and neither scenario is particularly satisfactory. Labidosaurikos is one of the stratigraphically oldest moradisaurines, but it is one of the phylogenetically youngest moradisaurines, at least according to the results of our weakly supported reduced phylogenetic analysis. Although the evidence does support an “out of equatorial Pangea” hypothesis, a more confident reconstruction of the pattern(s) of moradisaurine dispersals out of western Laurasia must await new fossil materials and more robust phylogenetic results.

Evolution of multiple rows of teeth in captorhinid reptiles

In recovering Rhodotheratus parvus as a moradisaurine, our phylogenetic results contrast with those of recent work on captorhinid interrelationships (Albright, Sumida & Jung, 2021; Matamales-Andreu et al., 2023), which proposed that multiple tooth rows (MTRs) evolved independently at least three times within Captorhinidae, i.e, in (1) Rhodotheratus parvus, (2) Captorhinus aguti, and (3) Moradisaurinae. Our work suggests that MTRs evolved only twice in this clade: in (1) C. aguti and (2) Moradisaurinae. The recovery of R. parvus as a moradisaurine is interesting, because its arrangement of MTRs (with apparently three rows on the maxilla, and two rows of teeth on the dentary: Albright, Sumida & Jung, 2021) is reminiscent of the pattern seen in C. aguti, in which the tooth rows are angled with respect to the main axis of the jaw, and which contrasts with the linear, parallel or subparallel organization of the MTRs in Tramuntanasaurus tiai and phylogenetically younger moradisaurines. The basal position of R. parvus in Moradisaurinae suggests the possibility that MTR pattern seen in T. tiai, Indosauriscus kuttyi, and their close moradisaurine relatives is derived from the condition seen in R. parvus. Previous studies of Captorhinus aguti dentition suggested a mechanism for the ontogenetic development of MTRs through differential, asymmetrical growth of the tooth-bearing maxillary and dentary bones, effectively moving the ontogenetically older teeth labially to such an extent that replacement teeth were able to attach to the alveolar shelf of the respective bones that were formed as the older teeth were displaced labially but not replaced by the next generation of teeth (Bolt & De Mar, 1975; de Ricqlès & Bolt, 1983; LeBlanc & Reisz, 2015). This mechanism allowed for the development of MTRs in Captorhinus aguti. Although this ontogenetic asymmetrical growth may explain the mechanism for this species, it is uncertain if this also applies to R. parvus. In addition, differences in the configuration of the MTRs in C. aguti and in most moradisaurines, together with the emplacement of MTRs on plate-like alveolar bone and evidence of tooth replacement (Modesto et al., 2019), are suggestive of different MTR development in moradisaurines exclusive of R. parvus. Histological analyses of the latter may help us resolve the apparent differences between the MTRs of most moradisaurines and those of C. aguti and R. parvus.

Paleobiology

Indosauriscus kuttyi is the only reptile known from the Kundarum Formation, the fauna of which is dominated numerically by dicynodont synapsids (Kutty, 1972; Ray, 1999) but includes medium-sized, indeterminate gorgonopsian synapsids (Ray & Bandyopadhyay, 2003). The dicynodonts range in size from small to relatively large herbivores (skull lengths [SLs] 50 mm to 350 mm; Ray & Bandyopadhyay, 2003). Indosauriscus kuttyi, with a skull length ranging 39 mm to 54 mm, falls at the small end of this spectrum of herbivore body size. Whereas the Kundarum captorhinid was described as omnivorous by Ray & Bandyopadhyay (2003), our identification of Indosauriscus kuttyi as a moradisaurine enriches the herbivore component of the Kundarum fauna.

In addition to its small size, Indosauriscus kuttyi is remarkable because it is one of the last representatives of an older Permian group of herbivorous reptiles that co-occurs with species of a geologically younger group of therapsid herbivores. Interestingly, the late Permian captorhinid Gansurhinus naobaogouensis is of interest because it is the only other moradisaurine that occurs with dicynodonts (Liu, 2023). Although approximately twice the size of the Kundarum moradisaurine, Gansurhinus naobaogouensis, with a maximum skull length of 110 mm (Liu, 2023), is, like Indosauriscus kuttyi, the smallest herbivore of its fauna (Naobaogou Formation, China; dicynodont SLs range 143 mm to 330 mm: Liu, 2021; Shi & Liu, 2023). Interestingly, Brocklehurst (2016) found that Gansurhinus qingtoushanensis (along with Captorhinikos valensis) was characterized by an extremely rapid decrease in an ancestral-size-reconstruction analysis of Captorhinidae. The observations that Gansurhinus naobaogouensis and Indosauriscus kuttyi are the smallest herbivores of their respective faunas contrast with that derived from Moradisaurus grandis, the largest and the last late Permian moradisaurine (SK of 400 mm: de Ricqlès & Taquet, 1982). Moradisaurus grandis occurs in an endemic fauna (Moradi Formation, Niger; Sidor et al., 2005) with only a medium-sized pareisaur as the co-occurring herbivore (Tsuji et al., 2013). Accordingly, the small body sizes of Indosauriscus kuttyi and Gansurhinus naobaogouensis in their respective faunas may have been an evolutionary response on the part of these reptiles to competition from the more numerous, and generally larger, dicynodonts. Testing this hypothesis will require more robust phylogenetic results for moradisaurines, as well as additional field work in the upper Permian continental strata.

Conclusions

We describe a new captorhinid reptile, Indosauriscus kuttyi, from the upper Permian Kundarum Formation of India. Phylogenetic analysis of captorhinids positions Indosauriscus kuttyi as an early branching moradisaurine. Indosauriscus kuttyi is the only herbivorous reptile in the Kundarum Formation, and it is the smallest herbivore in a terrestrial tetrapod fauna that is dominated by dicynodont therapsids. Its small body size may have been an adaptive response to competition with dicynodonts, a possibility hinted at by similar circumstances for Gansurhinus naobaogouensis in the Naobaogou Formation of China, but further data is needed to examine this hypothesis in detail.

Supplemental Information

Supplemental Information 1 Data matrix

A character list, explaining new characters and character modifications.

Supplemental Information 2 PAUP results based on the data matrix

We thank Christina Stoppa for the specimen illustrations, Diane Scott for the photographs, and the Indian Statistical Institute for loans, and colleagues Juan Cisneros and Michel Laurin for their reviews. We dedicate this work to the late Prof. T. S. Kutty who originally introduced this material to the scientific community.

Additional Information and Declarations

Competing Interests

Author Contributions

Data Availability

New Species Registration

The authors declare there are no competing interests.

Robert R. Reisz conceived and designed the experiments, performed the experiments, analyzed the data, prepared figures and/or tables, authored or reviewed drafts of the article, and approved the final draft.

Sankar Chatterjee analyzed the data, authored or reviewed drafts of the article, and approved the final draft.

Sean P. Modesto analyzed the data, prepared figures and/or tables, authored or reviewed drafts of the article, and approved the final draft.

The following information was supplied regarding data availability:

The data matrix employed in the phylogenetic analysis is available as a Supplemental File.

The specimens described are stored at the Indian Statistical Institute, Kolkata, India: ISIR 232, ISIR 233, ISIR 234, ISIR 235, and ISIR 236.

The following information was supplied regarding the registration of a newly described species:

Publication LSID: urn:lsid:zoobank.org:pub:47E6FA4C-AF21-40C3-8925-84B3CC374357

Genus name: urn:lsid:zoobank.org:act:DD1D0B12-FBE7-4A29-9EE7-2C5E772D8C01

Species name: urn:lsid:zoobank.org:act:853DC754-A6F3-4AE0-BE95-3C4C40334C36

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
