# Peer review of "A new moradisaurine captorhinid reptile (Amniota: Eureptilia) from the upper Permian of India"

_PeerJ, doi:10.7717/peerj.18394_

## Round 0.1 · original submission · Minor Revisions

Dear Dr. Reisz,

Your Ms # 97686 entitle “A new moradisaurine captorhinid reptile (Amniota: Eureptilia) from the upper Permian of India” has been reviewed by two reviewers. After analyzing their comments, I consider the Ms needs Minor Revisions before to be consider for publication in PeerJ.

I agree with the reviewers that the Ms is important and deserves publication. However, there are some minor issues to be consider before acceptance.
In this context, I consider that comments made by both reviewers on the phylogenetic analysis (and related missing reference) should be assess, particularly those related to the parameters used in the analysis. Please take particular attention to comments by Reviewer #1 in this matter.
Also, I agree with Reviewer #1 about to find a way to make clearer the paleobiogeographic discussion, which is an important goal of the general discussion of your paper. This could be done, as an alternative to the reviewer´s suggestions, by adding an additional figure (map) showing taxa distributions and modifying Figure 7 with a time calibrated analysis.

Thank you for submitting your Ms to PeerJ and I look forward to receiving your revision.

Sincerely,
Claudia Marsicano

·

Basic reporting

Reisz et al. ms, review

The descriptive portion seems fine, but I found some minor problems to solve before the paper can be published. Most importantly, the authors indicate that they used the data matrix from Sidor et al. 2022, but did not include this paper in the references! This is a rather serious omission because there are a number of minor (and one not so minor) problems with the phylogenetic analysis. The most serious problem is that the Nexus file provided in attachments indicates that all characters are unordered, but several have multiple states (3 to 5), and it seems likely that at least some of these are cline characters that should be ordered. This has been shown by the simulation of Rineau et al. (2015, 2018); ordering cline characters increases the proportion of correct clades found, and decreases the frequency of incorrect clades recovered. Unfortunately, I cannot provide more guidance because the state labels are not in the Mesquite file, and the Sidor et al. 2022 reference is not in the bibliography of the draft.

Other, more minor problems with the phylogenetic analysis or how its results are presented are:

On line 117, the authors declare that they performed both bootstrap and Bremer (decay) index analyses, but only some bootstrap values are reported in figure 7. Why?

Also, the legend of figure 7 explains the meaning of only part of the color coding: green, blue, and black. But there are also branches in red in fig. 7 part B. Why?

Last but not least, rather than presenting only those with >50% frequencies, I would present all those possible; PAUP allows showing all those >5%, I think.

For the geological age, the authors appear to be correct (Later Permian), but note that in addition to the Rubidge et al. 2013 paper, they might want to cite the more recent Smith et al. 2020 paper.

In the discussion of ontogenetic age, lines 190-198, the authors mention a premaxillary process of the nasal extending under the dorsal process of the premaxilla. They indicate that they observed it in mature individuals of a few captorhinid taxa, but not that this specimen is absent (and not worn off or broken off) in juvenile specimens, which is required for this character to be informative.

The description is fine, and the illustrations are very good, as usual for these authors.

In the discussion on evolution of body size (lines 503-506 and lines 568-584), I am surprised not to see Brocklehurst (2016, 2017) cited. They seem highly relevant.

The paleobiogeographic discussion is a bit hard to follow because no figure shows the tree with paleogeography mapped onto it. Yet, this is an easy exercise, which can be done in Mesquite, among many other computer programs. Ideally, this should be done on a timetree, so that the date of the dispersal events can also be assessed. For recent examples of how this can be done easily, the authors can look at (but need not cite!) Lemierre et al. (2024: fig. 10). Alternatively, a fancier, more rigorous method is available in other software, though I understand that the authors view the phylogeny as provisional and may not wish to invest the time required to do as in Longrich et al. (2020: fig. 10). Note that while the approach in this paper is very sophisticated, the timescale is not displayed properly, with only a numerical scale, without geological stages, which I find sloppy.

The sentence on lines 564-567 is sloppy; it implies that Ray & Brandyopadhyay (2003) decribed Indosauriscus kuttyi, which is erected in the draft! Modify the beginning of the sentence thus (or similar solutions): “Whereas the specimens now attributed to Indosauriscus kuttyi were described…

In the description of “Data availability”, what is a “conservative” Nexus format?

There are typographic errors:
Line 175 “Indosauriscus c”.
Line 514: “…basis for ‘out of…”
Line 516: “…, but there were also”
Line 530: “…at least according…”

As usual in my reports, I waive anonymity.

Best wishes

Michel Laurin

References:

Brocklehurst N (2016) Rates and modes of body size evolution in early carnivores and herbivores: a case study from Captorhinidae. PeerJ 4: e1555.

Brocklehurst N (2017) Rates of morphological evolution in Captorhinidae: An adaptive radiation of Permian herbivores. PeerJ 5:e3200.

Lemierre A, Bailon S, Folie A, Laurin M (2023) A new pipid from the Cretaceous of Africa (In Becetèn, Niger) and early evolution of the Pipidae. Journal of Systematic Palaeontology 21: 2266428. https://doi.org/10.1080/14772019.2023.2266428

Longrich NR, Suberbiola XP, Pyron RA, Jalil N-E (2020) The first duckbill dinosaur (Hadrosauridae: Lambeosaurinae) from Africa and the role of oceanic dispersal in dinosaur biogeography. Cretaceous Research 120: 104678.

Rineau V, Grand A, Zaragüeta R, Laurin M (2015) Experimental systematics: sensitivity of cladistic methods to polarization and character ordering schemes. Contributions to Zoology 84: 129-148. https://doi.org/10.1163/18759866-08402003

Rineau V, Zaragüeta I Bagils R, Laurin M (2018) Impact of errors on cladistic inference: simulation-based comparison between parsimony and three-taxon analysis. Contributions to Zoology 87: 25-40. https://doi.org/10.1163/18759866-08701003

Smith R, Rubidge B, Day M, Botha J (2020) Introduction to the tetrapod biozonation of the Karoo Supergroup. South African Journal of Geology 123: 131-140. https://doi.org/10.25131/sajg.123.0009

Experimental design

Not applicable (no experiments).

Validity of the findings

Mostly or all valid. See Basic reporting.

·

Basic reporting

No comment, the MS is well done.

Experimental design

The PAUP search parameters could be better described, so we can replicate these results in other programs. Which collapsing rule was used? How many trees per replicate? The paper that provided the data-matrix cited in the Methods section (Sidor et al. 2022) is no listed in the References.

Validity of the findings

no comments

Additional comments

There are only some typos, minor omissions and some problems with the figures (see attachment).

---

## Round 0.2 · Minor Revisions

Dear Sean,

Reviewing the new version of your Ms, I found minor things that need to be addressed before acceptance.

My comments are on the annotated Ms PDF and are related to the figures layout (3 & 5) and also to fig. 7´s legend, where, I think, there is an inconsistency in the text.

Please, submit the new version as soon as possible so I can continue processing your Ms.

Best regards
Claudia Marsicano

---

## Round 0.3 · accepted · Accept

Dear Dr. Modesto

I am pleased to inform you that your manuscript # 97686 entitled "A new moradisaurine captorhinid reptile (Amniota: Eureptilia) from the upper Permian of India", co-authored with Reisz and Chatterjee is now accepted for publication in PeerJ.

Thank you again for considering PeerJ and we look forward to your future contributions to the Journal.

sincerely,

Claudia Marsicano